# Designed on 0.18 μm CMOS Process Small Size Broadband Millimeter Wave Chip Antenna

**Ming-An Chung \***  **, Siao-Rong Huang and Pin-Rui Huang**

Department of Electronic Engineering, National Taipei University of Technology, Taipei 10608, Taiwan
* Correspondence: minganchung@ntut.edu.tw; Tel.: +886-2-2771-2171 (ext. 2212)

**Abstract:** This paper proposes a small-size broadband triangular monopole chip antenna for millimeter wave band applications. Process using 0.18 μm CMOS process and antenna design using Met-al_6. Triangular patch design and feed line length analysis to achieve a better reflection coefficient and also dig three circular slots at the grounding point to achieve better impedance matching. The operating frequency of the chip antenna is 62–100 GHz below the reflection coefficient −10 dB standard, with a fractional bandwidth of 54%. The maximum gain is −0.4 dBi at 64 GHz. The efficiency is 40.9%. The overall chip size is 1.2 × 1.2 (mm²). After measurement and verification, the proposed antenna reflection coefficient is similar to the simulation trend and has better resonance. The chip antenna frequency range proposed in this article covers the 5G NR FR2 frequency band. The proposed chip antenna can be applied in related fields such as the Internet of Things, Industry 4.0, and biomedical electronics.

**Keywords:** 5G; millimeter wave; chip antenna; CMOS

## 1. Introduction

The increasing demand for wireless connections has led to data traffic congestion on the traditional microwave spectrum. In response to such huge demand, the Third Generation Partnership Plan (3GPP) has expanded the frequency band of 5G New Radio (5G NR) to FR1 (0.4–7.1 GHz) and FR2 (24.2–52.6 GHz) to achieve higher connectivity, higher data rates, ultra-low latency, higher energy efficiency, and higher spectral efficiency to process these data and information [1]. Under the network coverage of the fifth-generation mobile communication, billions of physical devices worldwide are connected to the Internet and transmit a large amount of data and information through networks independent of human behavior [2,3]. The development of the Internet of Things has promoted the rapid growth of smart homes, smart cities, smart grids, industrial automation, and healthcare, and environment and traffic monitoring. Even if the novel coronavirus epidemic suddenly occurred in recent years, the demand for such a large number of equipment did not show a downward trend [4,5]. In recent years, with the transformation of mobile communication networks, more and more research has been conducted on the frequency bands of millimeter wave communication systems, including n257 (26.5–29.5 GHz), n258 (24.2–27.5 GHz), n259 (39.5–43.5 GHz), n260 (37–40 GHz), and n261 (27.5–28.3 GHz). Implementing high-performance, low-cost, and low-power transceivers has become an important research topic [6]. In addition to the above frequency bands, scientists are also actively studying the 60 GHz and 77 GHz communication frequency bands and applying them in the fields of industry, medical treatment, and automotive radar [7]. The sub-terahertz frequency band above 100 GHz is also attracting attention and will apply to physical systems, such as autonomous vehicles, robotic networks, virtual reality, or applications, such as intelligent sensing, imaging, defect detection, and short-range wireless communication [8,9].

In the past two decades, advanced process technology has led to a breakthrough in the growth of integrated circuits. Billions of transistors can be manufactured on a few

square millimeters of wafer and the precision of manufacturing technology is becoming increasingly mature [10]. Due to the improvement in the high-frequency characteristics of chip antennas through scaling, complementary metal oxide semiconductors (CMOS) have become a viable manufacturing method for chip antennas and can integrate a complete chip antenna system on a single chip through CMOS processes, further enhancing the competitiveness of the system [11,12]. However, the radiation efficiency and performance of chip antennas are still their disadvantages. In typical process of CMOS, the low resistivity and high dielectric coefficient of the silicon substrate will seriously cause the loss of the antenna and the silicon substrate will absorb most of electromagnetic waves, greatly reducing the radiation efficiency of the antenna [13]. However, compared to traditional antenna systems, chip antenna systems still have higher bandwidth, better integration advantages, and lower power consumption performance. In addition, as the most critical part of the system, the chip antenna eliminates parasitic effects and uncertainties on the circuit caused by additional connections. It can be directly connected to the circuit without requiring additional matching networks, providing designers with a more flexible design approach and facilitating chip layout [14,15]. Such a process has been applied in the frequency band of novel communication applications, such as millimeter wave and terahertz. The emergence of chip antennas has brought enormous potential to communication technology, playing an important role in the future Internet of Things [16,17].

In recent years, there has been extensive research in the field of millimeter wave chips, with different orientations and implementations. For example, resonance analysis [18–21], the introduction of artificial magnetic conductors [22,23], the design of integration with printed circuit boards [24,25], the design of integrated circuits [26], and miniaturization [27]. The influence of the number of twists and turns of a monopole antenna on the operating frequency is shown through simulation results. In reference [18], resonance is achieved at 28 GHz and 60 GHz, achieving the goal of dual frequency. In addition, considering the chip layout, a packaging substrate was selected and placed below the silicon substrate to generate constructive radiation at two frequencies, enhancing antenna gain and efficiency. Reference [19], coupled the feed antenna by adding a square resonant ring, which improved the gain of 7 dBi at 85 GHz and measured a maximum gain of 1.61 dB at 81.5 GHz. Reference [20] analyzed the dual ring antenna and analyzed the influence of dielectric resonator size on the radiation performance and gain of the antenna. In addition, a reflective plate has also been added to the design of the Yagi antenna, which has increased the gain by 0.8 dBi [21]. By conducting resonance analysis on the antenna, the best method to improve antenna performance can be determined. By adjusting the impedance and bandwidth through the use of parasitic capacitance generated by double-layer artificial magnetic conductors, it has been confirmed through measurement that the bandwidth has increased to 5.8 GHz and a consistent input impedance has been obtained [22]. Reference [23] analyzed and compared different artificial magnetic conductor units, and the results showed that the proposed (back-to-back E-shaped, B2BE) architecture had the best reflection coefficient (75–125 GHz) and was applied to antennas. With the assistance of artificial magnetic conductors, the gain and efficiency of chip antennas can be improved to a certain extent. However, in recent years, there have been studies and designs that integrate printed circuit boards and active circuits, which can overcome the disadvantages of low gain and low efficiency of chip antennas. For example, the use of air-filled substrate synthesis waveguide technology combined with chip antennas results in the better resonance of chip antenna radiation, with an efficiency improvement of over 90% [24] Alternatively, integrate the chip antenna with the active circuit and embed it into the printed circuit board to improve the overall antenna gain [25]. Reference [26] introduced a switching circuit in the chip antenna, which achieved an overall bandwidth of 53.4% in two operating states. Switching circuits can increase the operating mode of chip antennas and improve bandwidth performance. By using a hexagonal grid structure to reduce the size, [27] analyzed the angle of the grid to optimize impedance matching, and proposed a miniaturized broadband monopole antenna for the W-band. Additionally, the capacitance effect between the sixth layer of metal and

the first layer of metal was utilized to achieve broadband, with an antenna bandwidth of 31.5%. Chip antennas can be miniaturized using different architectures, and the capacitance effect between metal layers can be used to enhance bandwidth.

In this article, it is proposed that $1 \times 2$ broadband monopole chip antennae use 0.18 μm CMOS manufacturing process with an antenna area of $1.2 \times 1.2$ (mm²), which improves antenna radiation performance through array mode. The proposed triangular patch antenna design and circular slot holes enhance bandwidth performance. Measurements have verified that the bandwidth is 57.6–93.8 GHz, with a fractional bandwidth 54%, below the reflection coefficient −10 dB standard. Additionally, the proposed antenna has −0.4 dBi gain at 64 GHz. The antenna radiation efficiency is 40%. The frequency band covers the millimeter wave frequency band of 5G NR FR2. The explanation framework of this article is as follows: the second chapter explains the topology, simulation results, evolution, and parameter analysis of the proposed antenna. The third chapter compares the results of measurement and simulation to illustrate the differences between measurement and simulation. The fourth chapter describes the performance comparison with the recent literature and proposes the advantages of this antenna. Finally, in fifth chapter, the performance and characteristics of this antenna are summarized.

## 2. Materials and Methods

Figure 1 is a top view of the chip antenna proposed in this article, designed using a 0.18 μm CMOS process, which shown as Figure 1. This antenna composed of two triangular patches and a $1 \times 2$ microstrip line power divider. We used a GSG feed structure at the feed line, which is connected to Metal_1. For grounding, adding a circular groove design at the grounding point can achieve better impedance matching. The design of the trenching on the triangular patch in the picture is to comply with the limitations of the design rules. Figure 2 shows the 0.18 μm CMOS process stack architecture, with a silicon dioxide layer thickness of 10.34 μm, and a dielectric coefficient of 4. The silicon substrate thickness is 543.5 μm and the dielectric coefficient is 11.9. The triangular patch antenna and microstrip line power divider are implemented using a sixth metal layer.

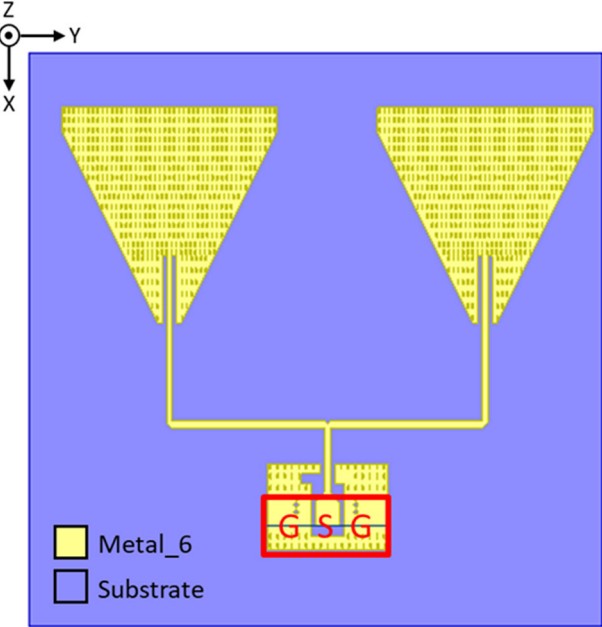

**Figure 1.** The top view of proposed chip antenna.

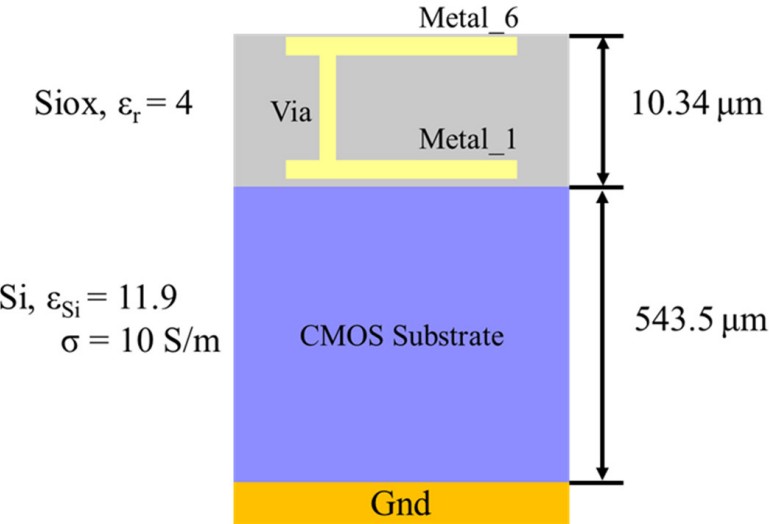

**Figure 2.** The antenna stack architecture of proposed chip antenna.

Figure 3 and the geometric diagram of the chip antenna proposed in this article. Table 1 shows the dimensions of the geometric parameters of each part of the chip antenna. The reflection coefficient, gain, and efficiency of the chip antenna are simulated, shown in Figures 4–6, respectively. The minimum reflection coefficient of chip antenna simulated at 64 GHz, with bandwidth of 57.6–93.8 GHz at the −10 dB standard. The maximum gain and efficiency are −0.4 dBi and 40.9%, respectively, at 64 GHz.

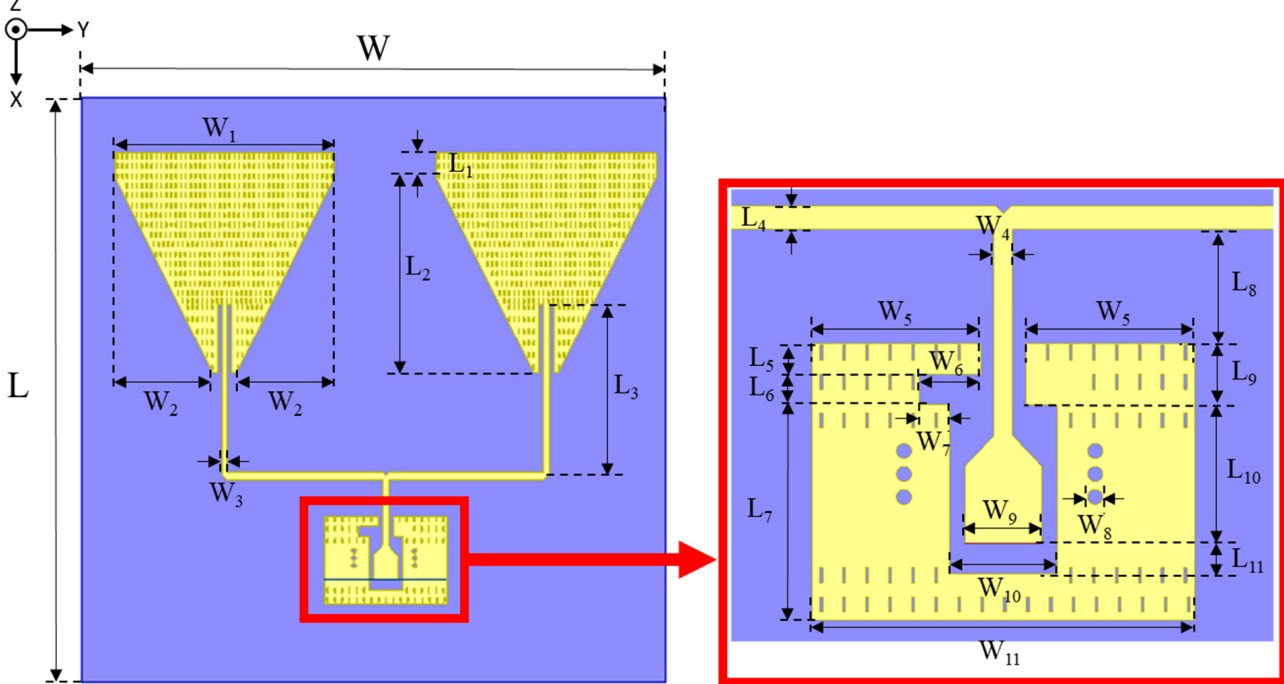

**Figure 3.** Proposed chip antenna geometry.

**Table 1.** Parameters of the proposed chip antenna.

| Parameter | Value (µm) | Parameter | Value (µm) |
|---|---|---|---|
| W | 1200 | L | 1200 |
| $W_1$ | 450 | $L_1$ | 50 |
| $W_2$ | 200 | $L_2$ | 400 |
| $W_3$ | 10 | $L_3$ | 350 |
| $W_4$ | 12 | $L_4$ | 15 |
| $W_5$ | 110 | $L_5$ | 20 |
| $W_6$ | 40 | $L_6$ | 20 |
| $W_7$ | 20 | $L_7$ | 140 |
| $W_8$ | 10 | $L_8$ | 75 |
| $W_9$ | 50 | $L_9$ | 45 |
| $W_{10}$ | 70 | $L_{10}$ | 90 |
| $W_{11}$ | 250 | $L_{11}$ | 20 |

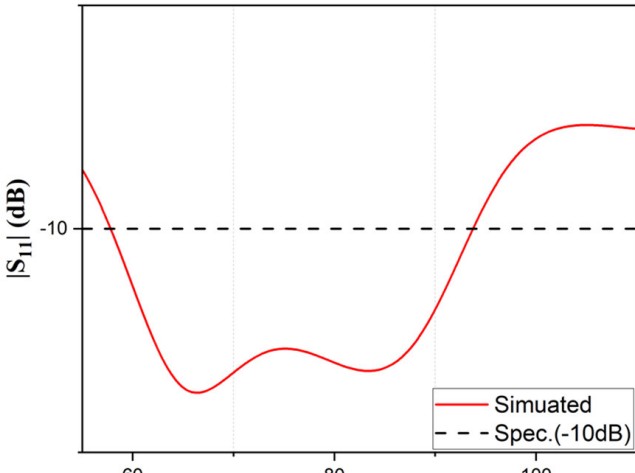

**Figure 4.** Simulated S11 of the antenna.

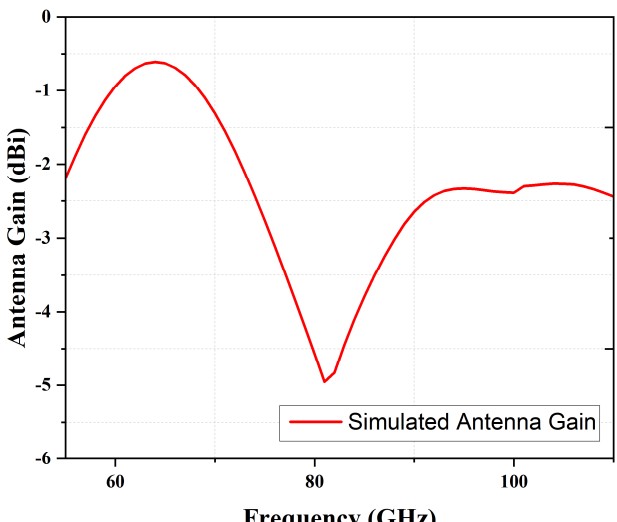

**Figure 5.** Simulated antenna gain.

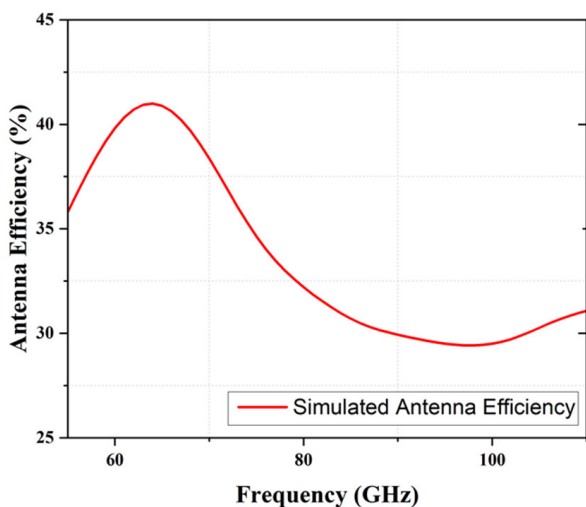

**Figure 6.** Simulated antenna efficiency.

## 2.1. Parameter Analysis of Antenna Feed Line

First, this paper analyzes the parameters of the feed line length $L_3$ for a single patch antenna. Figure 7 shows three different lengths of square monopole patch antennas, each 300 μm, 325 μm, and 350 μm, respectively. The results are shown in Figure 8, with the reflection coefficient of $-10$ dB as the bandwidth standard. When the feed line length $L_3$ is 300 μm, the bandwidth range is 88.6–94.5 GHz. When $L_3$ is 325 μm, the bandwidth range is 85.7–97.5 GHz. When the $L_3$ length is 350 μm, the bandwidth range is 83.1–100 GHz. From this parameter simulation, it can be seen that when the feed length is 350 μm, there is an optimal bandwidth range.

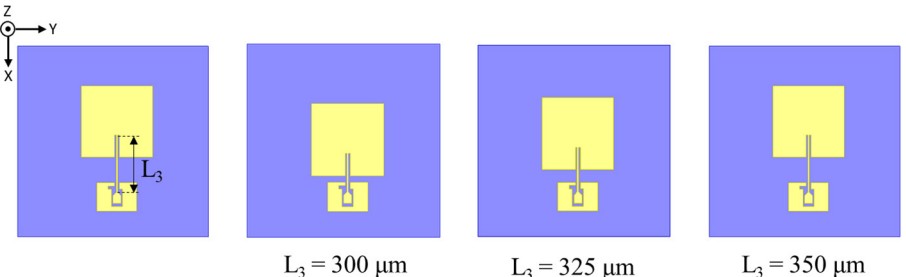

**Figure 7.** Antenna parameter $L_3$ evolution diagram.

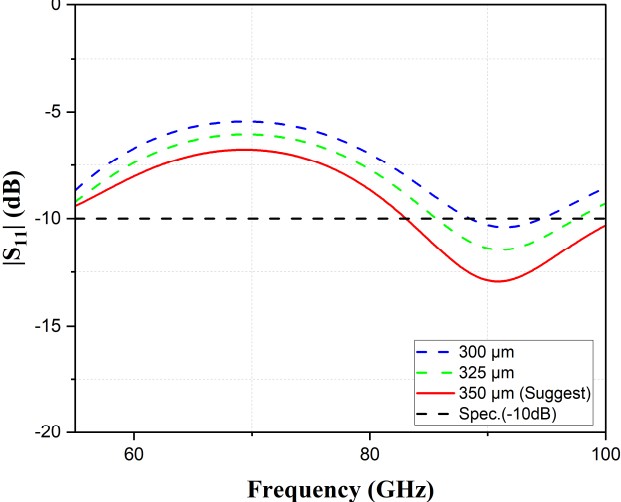

**Figure 8.** Reflection coefficient of antenna feed line variation.

### 2.2. Parameter Analysis of Triangular Patch

Next, we analyze the antenna parameter $W_2$, as shown in Figure 9, when $W_2$ is equal to 50, 125, and 200 μm, respectively. The simulation results are shown in Figure 10. When the reflection coefficient is −10 dB and the parameter $W_2$ is equal to 50 μm, the bandwidth range is 78.2–86.7 GHz. When $W_2$ is equal to 125 μm, the bandwidth range is from 75.5–87.7 GHz. When $W_2$ is equal to 200 μm, the bandwidth range is 73.8–89.1 GHz. From this parameter simulation, it can be seen that when the antenna parameter $W_2$ is equal to 200 μm, there is the best bandwidth performance.

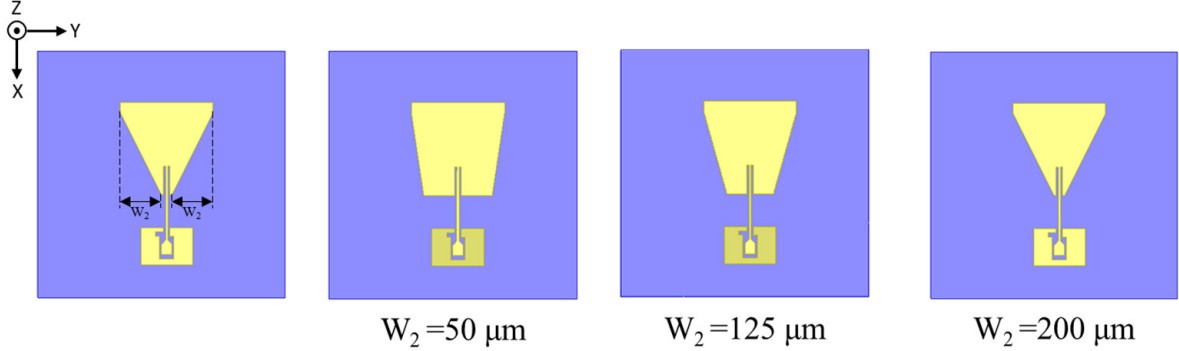

**Figure 9.** Triangular patch evolution diagram.

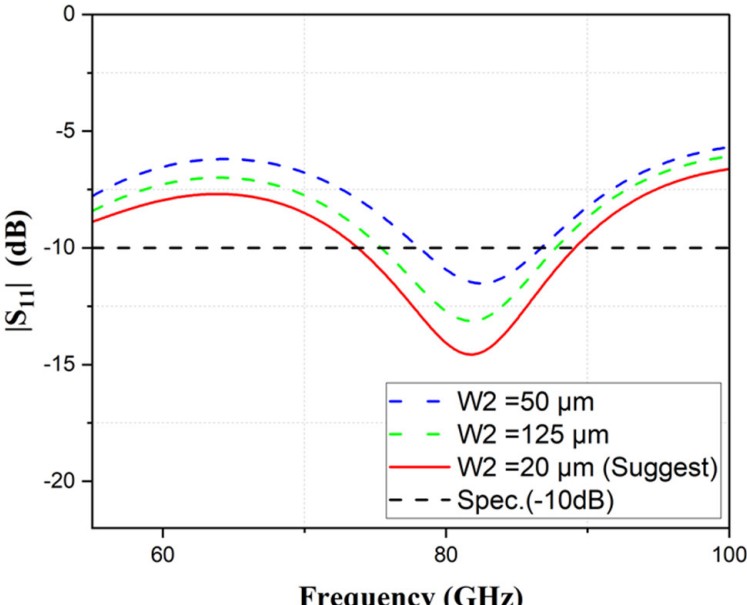

**Figure 10.** Reflection coefficient of $W_2$ variation.

### 2.3. Parameter Analysis of Different Numbers of Circular Slot

In this article, the original triangular patch antenna is transmitted through $1 \times 2$ power divider of 2 expands it into a $1 \times 2$ array antenna and performs parameter analysis by digging a circular slot in the ground. As shown in Figure 11, 1, 2, and 3, circular slot holes were dug at the ground, and the bandwidth range was observed at a reflection coefficient of −10 dB. The results are shown in Figure 12. When digging a circular slot hole, the bandwidth range is between 62.2 GHz and 88.1 GHz. When digging two circular slots, the bandwidth range is 59.6–91.7 GHz. When digging three circular slots, the bandwidth range is from 57.9–93.7 GHz. From the simulation results, it can be seen that the optimal bandwidth range is achieved when three circular slots are dug at the grounding point.

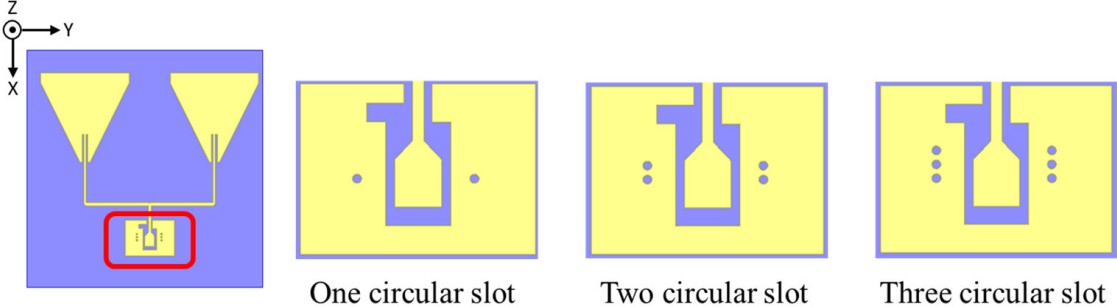

**Figure 11.** Diagram of different numbers of circular slot.

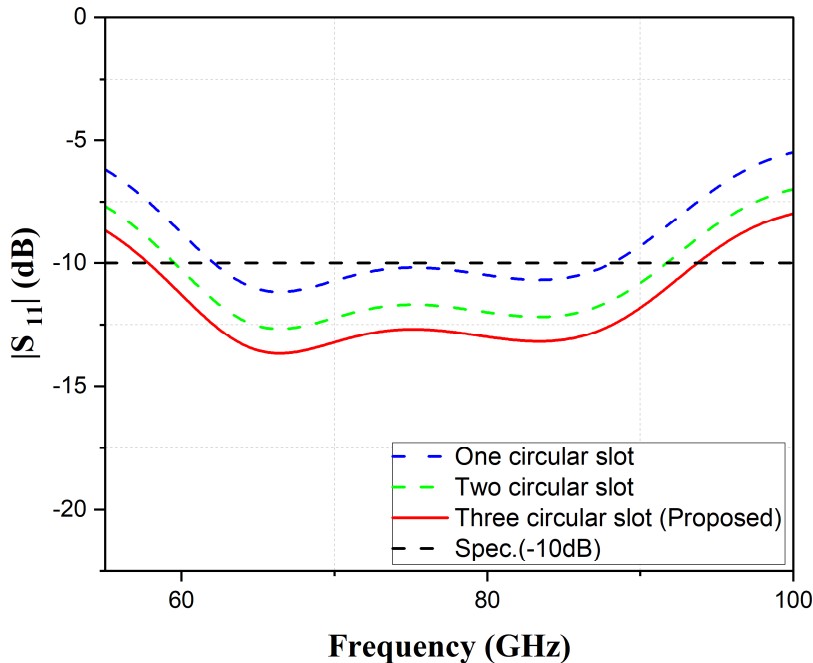

**Figure 12.** Reflection coefficient of different numbers of circular slot.

*2.4. Surface Current Analysis*

The Figure 13 shows the current distribution of the antenna at 64 GHz. The chip antenna proposed in this article uses a GSG probe to feed. The current flows evenly through the feed line and power divided into two triangular patch antennas. The current is mainly distributed on both sides of the power divider and triangular patch antenna. As can be seen from the figure, the current distribution of the power divider is average and consistent. After the triangular patch antenna is connected, the current is mainly distributed on both sides of the triangular patch antenna, making the antennas at both ends generate optimal resonance.

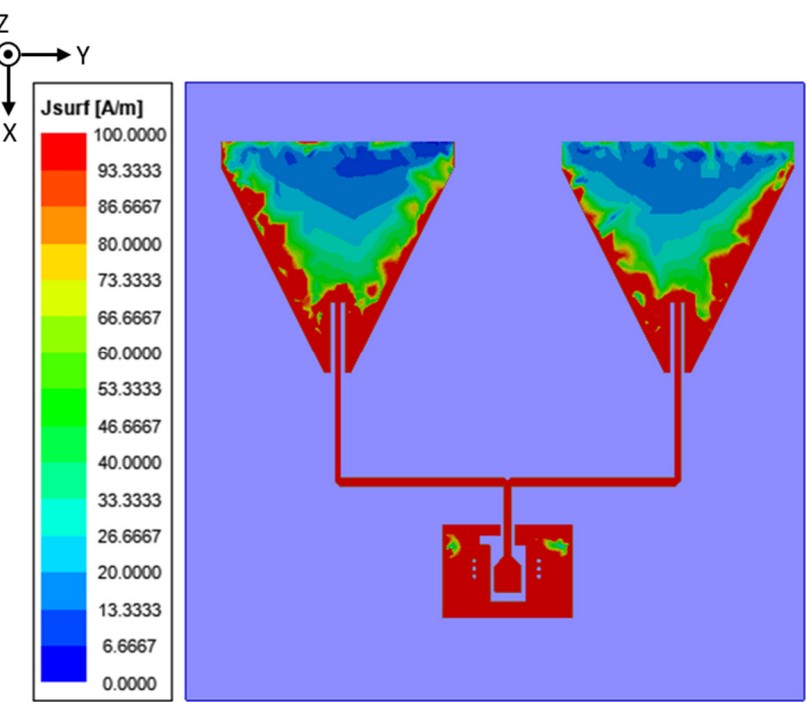

**Figure 13.** An amount of 64 GHz surface current of chip antenna.

## 3. Results

In order to verify the design results, we measured the reflection coefficient of the antenna. Figure 14 shows the antenna measurement state diagram taken using a high magnification lens. The instrument for this measurement is shown in Figure 15. The Agilent E7350A vector network analyzer is used for measurement, with a measurement range of 2 GHz to 110 GHz. Using a GSG probe feed to measure the reflection coefficient, the probe spacing is 50 μm. As shown in Figure 16, the measured operating frequency is 62–100 GHz at the −10 dB reflection coefficient standard. The trend of simulation and measurement results is generally similar, but the measured results have a better reflection coefficient compared to the simulation, presumably due to the impedance matching effect when the probe is inserted.

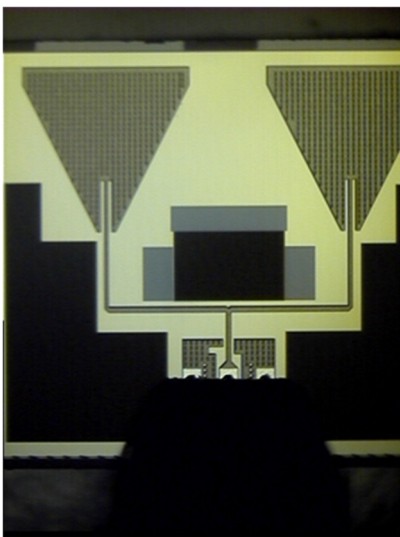

**Figure 14.** GSG connector and actual measurement diagram.

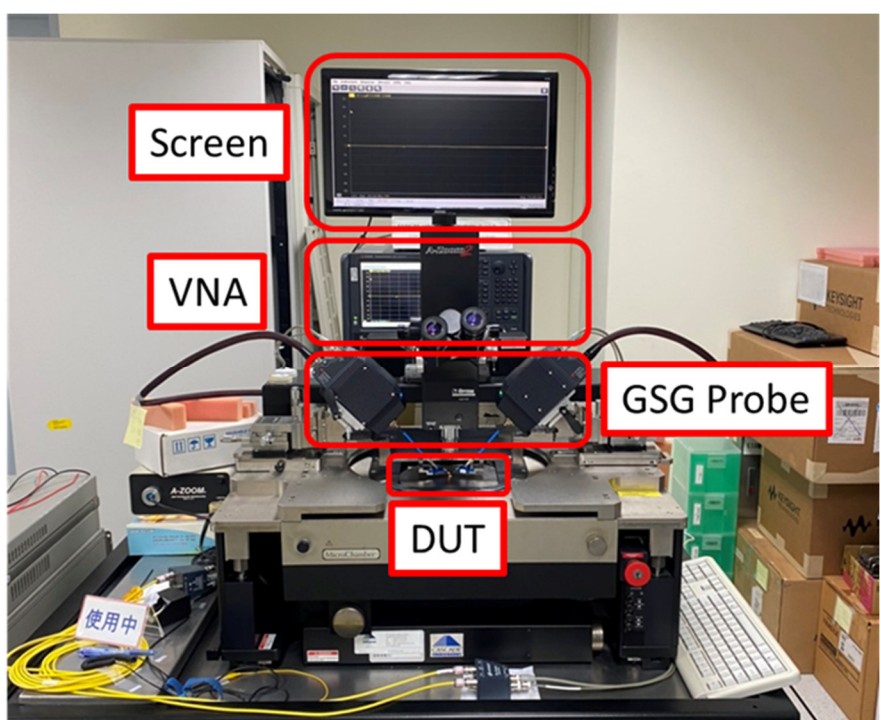

**Figure 15.** Measurement environment setup.

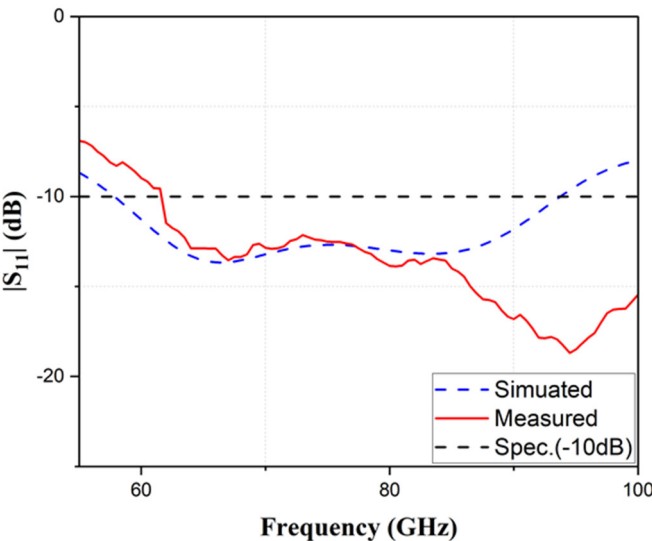

**Figure 16.** Reflection coefficient of simulation and measurement of chip antenna.

Figure 17a,b illustrates chip antenna E plane and H plane views at 64 GHz. As shown in Figure 17a, at an operating frequency of 64 GHz, the E plane field type has a maximum gain of −0.4 dBi at a Theta of 0 deg. Figure 17b shows the radiation field pattern of the antenna in the H plane, with a maximum gain of −0.4 dBi when Phi is 0 deg. From the above results, it can be seen that the chip antenna proposed in this article radiates upward and has a good radiation field pattern.

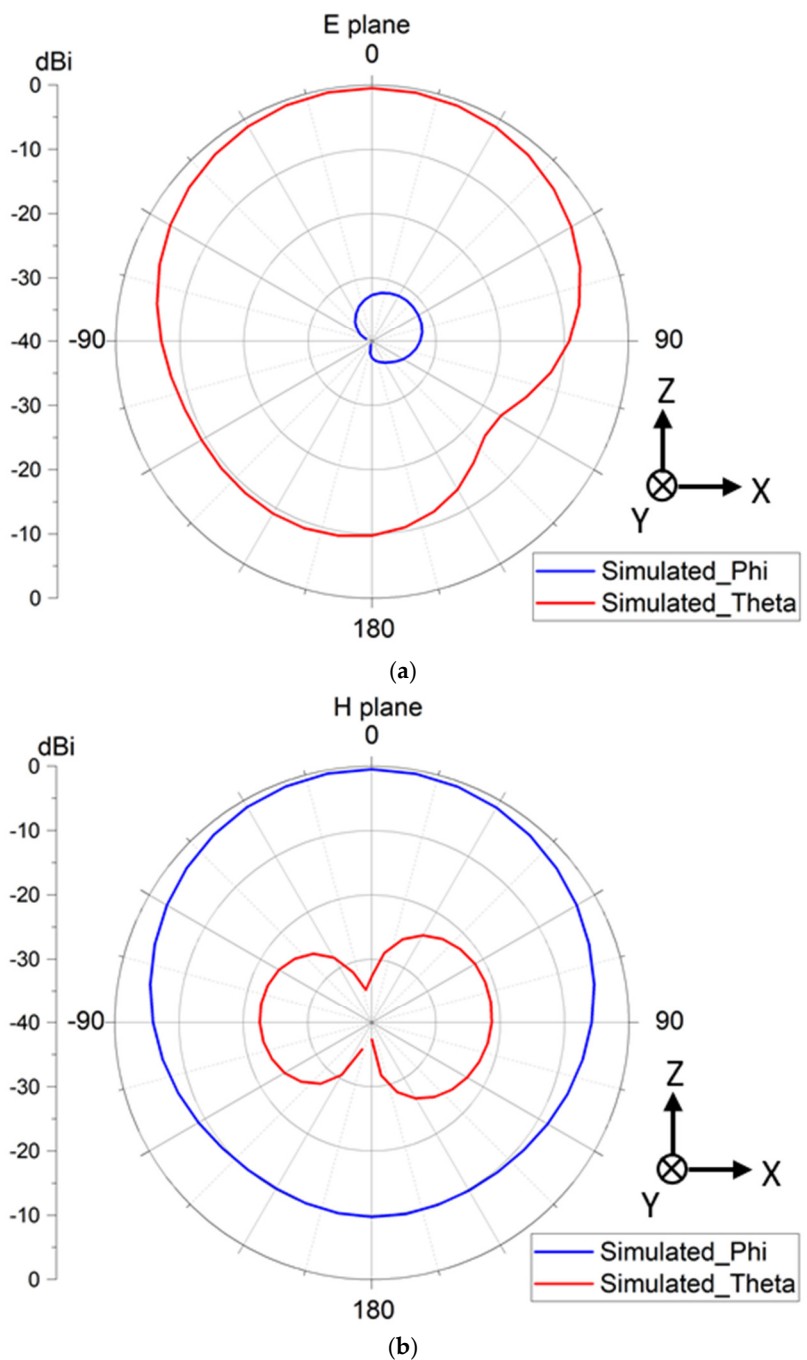

**Figure 17.** 2D Radiation Pattern (**a**) E Plane; (**b**) H Plane.

## 4. Discussion

From the above parameter analysis, it can be seen that designing the feed line length of the patch antenna and the triangular patch antenna can help achieve a better reflection coefficient and frequency bandwidth percentage. Additionally, better impedance matching can be achieved by projecting a frog circular slot in both directions. Table 2 provides various parameters related to the chip antenna literature in recent years. The process, frequency, bandwidth, gain, efficiency, and size used in each document are listed in the table. Compared with the chip antenna design proposed in this article, it can be seen that the bandwidth is better than that of the references [18–27]. In terms of gain, it is better than references [18,20,22,23]. The efficiency is better than the literature [18,20,22,25]. The dimensions have advantages over references [19–21,26]. Based on the above, the chip

antenna proposed in this paper is competitive in terms of bandwidth, gain, efficiency, and size. The antenna proposed in this paper has good bandwidth and radiation performance and is suitable for application in millimeter wave systems.

**Table 2.** Comparison table with the other research literature.

| Reference | Process | * Freq. (GHz) | * FBW (%) | Gain (dBi) | * Eff. (%) | Size (mm$^2$) |
|---|---|---|---|---|---|---|
| This Work | 0.18 μm CMOS | 64 | 54 | −0.4 | 40.9 | 1.2 × 1.2 |
| [18] | 0.65 μm CMOS | 28, 60 | 5.3, 5.9 | −10, 4 | 45, 30 | 0.25 × 0.3 |
| [19] | 0.13 μm SiGe CMOS | 81.5 | 12.1 | 1.61 | NA | 1.296 × 1.508 |
| [20] | 0.65 μm CMOS | 24, 40 | 19, 20 | −1, 0 | 41, 31 | 2.5 × 2.5 × 2.5 |
| [21] | 0.18 μm CMOS | 60 | 16 | 0.3 | 45 | 5 × 5 |
| [22] | CMOS | 60 | 9.8 | −2.3 | 21 | 1.5 × 0.75 |
| [23] | 0.13 μm BiCMOS | 84 | 11.7 | −0.5 | 41 | 0.32 × 0.1 |
| [24] | MEMS | 28.5 | 13 | 6.6 | 90.9 | 0.52 × 0.52 |
| [25] | 0.28 μm CMOS | 33 | 16.3 | 14 | 37.3 | 0.66 × 0.85 |
| [26] | 0.18 μm CMOS | 40 | 53.4 | 3.3 | NA | 1.1 × 1.7 |
| [27] | 0.13 μm CMOS | 81 | 31.4 | −0.3 | NA | 0.1 × 0.05 |

* Freq.: Frequency; * FBW: Fractional bandwidth; * Eff: Efficiency.

## 5. Conclusions

Chip antennas are a good candidate for millimeter wave system applications that require high integration and broadband characteristics. This paper proposes a chip antenna suitable for millimeter wave systems using 0.18 μm CMOS process, which size is 1.2 × 1.2 (mm$^2$), using an M6 metal layer for antenna design. Through parameter analysis, it can be seen that the optimization of the feed line length and patch shape is helpful for antenna impedance matching. In addition, by expanding to a 1 × 2 array antenna pole and digging three circular slots in the ground, antenna radiation performance is increased. The proposed chip antenna in this paper has a gain of −0.4 dBi at 64 GHz, maximum. The proposed antenna has a fractional bandwidth 54%. The actual measured bandwidth is 62–100 GHz, and the analog and measured bandwidth are similar. The chip antenna frequency range proposed in this article covers the 5G NR FR2 frequency band and can be used in the fields of the Internet of Things, Industry 4.0, and biomedical electronics. Its small size makes it easy to integrate with other microwave circuits, which can form a highly integrated millimeter wave system. In the future, it will be able to integrate with microwave circuits or combine applications with printed circuit boards to further improve antenna gain performance.

**Author Contributions:** Conceptualization, M.-A.C.; methodology, M.-A.C.; software, M.-A.C. and S.-R.H.; validation, M.-A.C., S.-R.H. and P.-R.H.; formal analysis, M.-A.C., S.-R.H. and P.-R.H.; investigation, M.-A.C., S.-R.H. and P.-R.H.; resources, M.-A.C.; writing—original draft preparation, M.-A.C. and S.-R.H.; writing—review and editing, M.-A.C.; visualization, M.-A.C.; supervision, M.-A.C.; project administration, M.-A.C.; funding acquisition, M.-A.C. All authors have read and agreed to the published version of the manuscript.

**Funding:** This research received no external funding.

**Data Availability Statement:** All data are included within the manuscript.

**Acknowledgments:** Not applicable.

**Conflicts of Interest:** The authors declare no conflict of interest.

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
