# Peer review of "Designed on 0.18 μm CMOS Process Small Size Broadband Millimeter Wave Chip Antenna"

_inventions, doi:10.3390/inventions8030064_

Round 1

Reviewer 1 Report

The manuscript demonstrated small size broadband millimeter wave chip antenna based on CMOS process. The manuscript is well organized. There are some points to be revised.

1.     In introduction, the authors use ‘Literature analyzed’ and ‘in literature’ repeatly. More specific and diversified statement would help understanding.

2.     The parameters of thickness should be added to Figure 2, as described in Section 2.

Reviewer 2 Report

This article proposes a small size broadband triangular monopole chip antenna for 7mm waveband applications. Which is a novel invention that can be applied in various industries and promote the use of the internet of things.

Reviewer 3 Report

Authors have presented their idea and prototype in a concise manner. Few improvements can be made to improve the quality of the journal.

1.In Lines 105 to 110, authors have mentioned as chapters and also the points mentioned does not match with the section headings, all that lines need to be checked and corrected. 

2.Author is suggested to give a brief summary from the survey and also highlight their work based on the survey.

3. Result and Discussion section has only qualitative contents related to simulation. Author is suggested to give a quantitative view of the work based on the related works considered. 
